# ADVERSARIALLY LEARNED MIXTURE MODEL

## ABSTRACT

The Adversarially Learned Mixture Model (AMM) is a generative model for unsupervised or semi-supervised data clustering. The AMM is the first adversarially optimized method to model the conditional dependence between inferred continuous and categorical latent variables. Experiments on the MNIST and SVHN datasets show that the AMM allows for semantic separation of complex data when little or no labeled data is available. The AMM achieves unsupervised clustering error rates of 3.32% and 20.4% on the MNIST and SVHN datasets, respectively. A semi-supervised extension of the AMM achieves a classification error rate of 5.60% on the SVHN dataset.

## 1 INTRODUCTION

Semi-supervised or unsupervised representation learning enables the utilization of all available data when tackling problems where there are little or no labeled examples. This is a common scenario in many applications of machine learning, such as medical image analysis, where it is reinforced by the expense of obtaining expert labeled examples. Moreover, machine-learned representations are more likely to be used for subsequent tasks if they are interpretable and meaningful. Deep generative modelling is a suitable approach to this problem, as derived models have been shown capable of learning from both labeled and unlabeled examples, embedding data according to desired latent variable distributions, and producing realistic data examples generated from samples of those latent variables.

The Generative Adversarial Network (GAN) has recently emerged as a powerful framework for modeling complex data distributions without having to approximate intractable likelihoods. In the formulation by Goodfellow et al. (2014), a GAN consists of two networks: a *generator $G$* that is trained to yield unique samples from the data distribution, and a *discriminator $D$* that is trained to distinguish between generated and true data samples.

Dumoulin et al. (2016) and Donahue et al. (2017) have proposed the ALI and BiGAN models that add an inference process, i.e., the ability to map data samples to points in the latent space, to the GAN framework. A second generator for inference, or *encoder*, is added to the original GAN generator and the discriminator is adapted for the two-dimensional space of data inputs and latent representations. A variant of the resulting model is also introduced by Dumoulin et al. (2016) for conditional data generation, but still assumes that the class of the data is always observed, as inference of categorical variables is not included.

Adversarial approaches for the inference of both continuous and categorical variables are actively researched. Chen et al. (2016) introduce a hybrid adversarial method that is capable of modelling both continuous and categorical latent variables for unsupervised clustering and feature disentanglement. Another hybrid adversarial method is introduced by Makhzani et al. (2016) where adversarial objectives on continuous and categorical latent variables are optimized for unlabeled examples and categorical cross entropy on categorical variables is optimized for labeled examples. Li et al. (2017) and Deng et al. (2017) point toward fully adversarial semi-supervised classification using inferred categorical variables by introducing a "three player" adversarial game, but stop short by adding auxiliary "collaborative" objectives. In each of these methods, it is assumed that categorical and continuous latent variables are independently distributed. This independence assumption results in discontinuities in the latent space between categories, which removes the notion of inter-categorical proximity.

Another notable family of generative models, Variational Autoencoders (VAEs), maximize the posterior distribution of latent representations given the data instead of using an adversarial approach. As VAEs integrate inference, semi-supervised classification can be performed by conditioning the

continuous latent variable of the VAE on the class label (Kingma et al., 2014; Dilokthanakul et al., 2016; Maaløe et al., 2017). However, the quality of VAE results depend on the expressiveness of the inference distribution and every time the assumptions about the inference or data distributions are changed a new objective function needs to be derived. In this way, variational optimization is not as versatile as adversarial training.

We present the Adversarially Learned Mixture Model (AMM). The AMM is, to our knowledge, the first generative model inferring both continuous and categorical latent variables to perform either unsupervised or semi-supervised clustering of data using a single adversarial objective. This is enabled, in part, by explicitly modelling the dependence between continuous and categorical latent variables, which eliminates discontinuities between categories in the latent space. Semi-supervised clustering and classification is enabled by a simplified formulation of the "three player game", presented by Li et al. (2017). In this paper we show that the AMM achieves an unsupervised clustering error rate of 3.32% and 20.4% on the MNIST (LeCun & Cortes, 2010) and SVHN datasets (Netzer et al., 2011) respectively, and that a semi-supervised extension, SAMM, achieves a classification error rate of 5.60% on the SVHN dataset . To support the reproducibility of the methodology and experiments presented, a public version of the code will be made available.

## 2 METHOD

### 2.1 PRELIMINARIES

The ALI and BiGAN models are trained by matching two joint distributions of images $\boldsymbol{x} \in \mathbb{R}^D$ and their latent code $\boldsymbol{z} \in \mathbb{R}^L$. The two distributions to be matched are the inference distribution $q(\boldsymbol{x},\boldsymbol{z})$ and the synthesis distribution $p(\boldsymbol{x},\boldsymbol{z})$, where,

$$q(\boldsymbol{x},\boldsymbol{z}) = q(\boldsymbol{x})q(\boldsymbol{z}\,|\,\boldsymbol{x}), \qquad (1)$$
$$p(\boldsymbol{x},\boldsymbol{z}) = p(\boldsymbol{z})p(\boldsymbol{x}\,|\,\boldsymbol{z}). \qquad (2)$$

Samples of $q(\boldsymbol{x})$ are drawn from the training data and samples of $p(\boldsymbol{z})$ are drawn from a prior distribution, usually $\mathcal{N}(0,1)$. Samples from $q(\boldsymbol{z}\,|\,\boldsymbol{x})$ and $p(\boldsymbol{x}\,|\,\boldsymbol{z})$ are drawn from neural networks that are optimized during training. Dumoulin et al. (2016) show that sampling from $q(\boldsymbol{z}\,|\,\boldsymbol{x}) = \mathcal{N}(\mu(\boldsymbol{x}),\sigma^2(\boldsymbol{x})I)$ is possible by employing the reparametrization trick (Kingma & Welling, 2013), i.e. computing

$$\boldsymbol{z} = \mu(\boldsymbol{x}) + \sigma(\boldsymbol{x}) \odot \epsilon, \quad \epsilon \sim \mathcal{N}(0,I), \qquad (3)$$

where $\odot$ is element wise vector multiplication.

A conditional variant of ALI has also been explored by Dumoulin et al. (2016) where an observed class-conditional categorical variable $\boldsymbol{y}$ has been introduced. The joint factorization of each distribution to be matched are:

$$q(\boldsymbol{x},\boldsymbol{y},\boldsymbol{z}) = q(\boldsymbol{x},\boldsymbol{y})q(\boldsymbol{z}\,|\,\boldsymbol{y},\boldsymbol{x}), \qquad (4)$$
$$p(\boldsymbol{x},\boldsymbol{y},\boldsymbol{z}) = p(\boldsymbol{y})p(\boldsymbol{z})p(\boldsymbol{x}\,|\,\boldsymbol{y},\boldsymbol{z}). \qquad (5)$$

Samples of $q(\boldsymbol{x},\boldsymbol{y})$ are drawn from the data. Samples of $p(\boldsymbol{z})$ are drawn from a continuous prior on $\boldsymbol{z}$, and samples of $p(\boldsymbol{y})$ are drawn from a categorical prior on $\boldsymbol{y}$, both of which are marginally independent. Samples from $q(\boldsymbol{z}\,|\,\boldsymbol{y},\boldsymbol{x})$ and $p(\boldsymbol{x}\,|\,\boldsymbol{y},\boldsymbol{z})$ are drawn from neural networks that are optimized during training.

In the following sections we present graphical models for $q(\boldsymbol{x},\boldsymbol{y},\boldsymbol{z})$ and $p(\boldsymbol{x},\boldsymbol{y},\boldsymbol{z})$ that build off of conditional ALI. Where conditional ALI requires the full observation of categorical variables, the models we present will account for both unobserved and partially observed categorical variables. We finally show how they can be optimized using a single adversarial objective.

### 2.2 ADVERSARIALLY LEARNED MIXTURE MODEL

The AMM is an adversarial generative model for deep unsupervised clustering of data. Figure 1 presents an overview of the model.

Like conditional ALI, a categorical variable is introduced to model the labels. However, the unsupervised setting now requires a different factorization of the inference distribution in order to

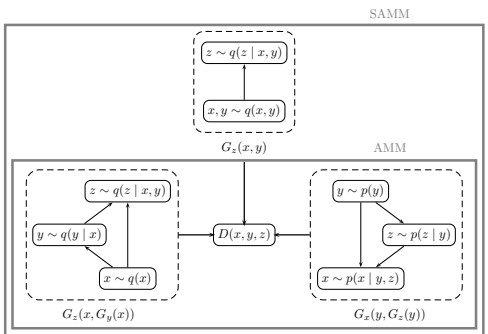

Figure 1: Overview of the unsupervised (AMM) and semi-supervised (SAMM) model with the first option (Equation (6)) for the inference distribution. AMM consists of two generators, encoder $G_{\boldsymbol{z}}(\boldsymbol{x}, G_{\boldsymbol{y}}(\boldsymbol{x}))$ and decoder $G_{\boldsymbol{x}}(\boldsymbol{y}, \boldsymbol{z})$, and a discriminator $D(\boldsymbol{x}, \boldsymbol{y}, \boldsymbol{z})$. SAMM includes an additional generator for labeled data, $G_z(\boldsymbol{x}, \boldsymbol{y})$.

enable inference of the categorical variable $\boldsymbol{y}$, namely:

$$q_1(\boldsymbol{x}, \boldsymbol{y}, \boldsymbol{z}) = q(\boldsymbol{x})q(\boldsymbol{y} \,|\, \boldsymbol{x})q(\boldsymbol{z} \,|\, \boldsymbol{x}, \boldsymbol{y}), \tag{6}$$

or

$$q_2(\boldsymbol{x}, \boldsymbol{y}, \boldsymbol{z}) = q(\boldsymbol{x})q(\boldsymbol{z} \,|\, \boldsymbol{x})q(\boldsymbol{y} \,|\, \boldsymbol{x}, \boldsymbol{z}). \tag{7}$$

Samples of $q(\boldsymbol{x})$ are drawn from the training data, and samples from $q(\boldsymbol{y} \,|\, \boldsymbol{x})$, $q(\boldsymbol{z} \,|\, \boldsymbol{x}, \boldsymbol{y})$ or $q(\boldsymbol{z} \,|\, \boldsymbol{x})$, $q(\boldsymbol{y} \,|\, \boldsymbol{x}, \boldsymbol{z})$ are generated by neural networks. The reparametrization trick is not directly applicable to discrete variables and multiple methodologies have been introduced to approximate categorical samples (Jang et al., 2016; Maddison et al., 2017). We follow Kendall & Gal (2017) and sample from $q(\boldsymbol{y} \,|\, \boldsymbol{x})$ by computing

$$
\begin{aligned}
h_{\boldsymbol{y}}(\boldsymbol{x}) &= \mu_{\boldsymbol{y}}(\boldsymbol{x}) + \sigma_{\boldsymbol{y}}(\boldsymbol{x}) \odot \epsilon, \quad \epsilon \sim \mathcal{N}(0, I), \tag{8} \\
y(\boldsymbol{x}) &= \mathrm{softmax}(h_{\boldsymbol{y}}(\boldsymbol{x})). \tag{9}
\end{aligned}
$$

Then, we can sample from $q(\boldsymbol{z} \,|\, \boldsymbol{x}, \boldsymbol{y})$ by computing

$$z(\boldsymbol{x}, h_{\boldsymbol{y}}(\boldsymbol{x})) = \mu_{\boldsymbol{z}}(\boldsymbol{x}, h_{\boldsymbol{y}}(\boldsymbol{x})) + \sigma_{\boldsymbol{z}}(\boldsymbol{x}, h_{\boldsymbol{y}}(\boldsymbol{x})) \odot \epsilon, \quad \epsilon \sim \mathcal{N}(0, I). \tag{10}$$

A similar sampling strategy can be used to sample from $q(\boldsymbol{y} \,|\, \boldsymbol{x}, \boldsymbol{z})$ in (7).

The factorization of the synthesis distribution $p(\boldsymbol{x}, \boldsymbol{y}, \boldsymbol{z})$ also differs from conditional ALI:

$$p(\boldsymbol{x}, \boldsymbol{y}, \boldsymbol{z}) = p(\boldsymbol{y})p(\boldsymbol{z} \,|\, \boldsymbol{y})p(\boldsymbol{x} \,|\, \boldsymbol{y}, \boldsymbol{z}). \tag{11}$$

The product $p(\boldsymbol{y})p(\boldsymbol{z} \,|\, \boldsymbol{y})$ can be conveniently given by a mixture model. Samples from $p(\boldsymbol{y})$ are drawn from a multinomial prior, and samples from $p(\boldsymbol{z} \,|\, \boldsymbol{y})$ are drawn from a continuous prior, for example, $\mathcal{N}(\mu_{\boldsymbol{y}=k}, 1)$. Samples from $p(\boldsymbol{z} \,|\, \boldsymbol{y})$ can alternatively be generated by a neural network by again employing the reparameterization trick. Namely,

$$z(\boldsymbol{y}) = \mu(\boldsymbol{y}) + \sigma(\boldsymbol{y}) \odot \epsilon, \quad \epsilon \sim \mathcal{N}(0, I). \tag{12}$$

This approach effectively learns the parameters of $\mathcal{N}(\mu_{\boldsymbol{y}=k}, \sigma_{\boldsymbol{y}=k})$.

### 2.2.1 ADVERSARIAL VALUE FUNCTION

We follow Dumoulin et al. (2016) and define the value function that describes the unsupervised game between the discriminator $D$ and the generator $G$ as:

$$
\begin{aligned}
\min_{G} \max_{D} V(D, G) &= \mathbb{E}_{q(\boldsymbol{x})}[\log(D(\boldsymbol{x}, G_{\boldsymbol{y}}(\boldsymbol{x}), G_{\boldsymbol{z}}(\boldsymbol{x}, G_{\boldsymbol{y}}(\boldsymbol{x}))))] \\
&\quad + \mathbb{E}_{p(\boldsymbol{y}, \boldsymbol{z})}[\log(1 - D(G_{\boldsymbol{x}}(\boldsymbol{y}, G_{\boldsymbol{z}}(\boldsymbol{y})), \boldsymbol{y}, G_{\boldsymbol{z}}(\boldsymbol{y})))] \\
&= \iiint q(\boldsymbol{x})q(\boldsymbol{y} \,|\, \boldsymbol{x})q(\boldsymbol{z} \,|\, \boldsymbol{x}, \boldsymbol{y})\log(D(\boldsymbol{x}, \boldsymbol{y}, \boldsymbol{z}))d\boldsymbol{x}d\boldsymbol{y}d\boldsymbol{z} \\
&\quad + \iiint p(\boldsymbol{y})p(\boldsymbol{z} \,|\, \boldsymbol{y})p(\boldsymbol{x} \,|\, \boldsymbol{y}, \boldsymbol{z})\log(1 - D(\boldsymbol{x}, \boldsymbol{y}, \boldsymbol{z}))d\boldsymbol{x}d\boldsymbol{y}d\boldsymbol{z}.
\end{aligned}
\tag{13}
$$

There are four generators in total: two for the encoder $G_{\boldsymbol{y}}(\boldsymbol{x})$ and $G_{\boldsymbol{z}}(\boldsymbol{x},G_{\boldsymbol{y}}(\boldsymbol{x}))$, which map the data samples to the latent space; and two for the decoder $G_{\boldsymbol{z}}(\boldsymbol{y})$ and $G_{\boldsymbol{x}}(\boldsymbol{y},G_{\boldsymbol{z}}(\boldsymbol{y}))$, which map samples from the prior to the input space. $G_{\boldsymbol{z}}(\boldsymbol{y})$ can either be a learned function, or be specified by a known prior (see Algorithm 1 for a detailed description of the optimization procedure).

---

**Algorithm 1** AMM training procedure using distributions (6) and (11).

---

$\theta_{G_{\boldsymbol{y}}(\boldsymbol{x})},\theta_{G_{\boldsymbol{z}}(\boldsymbol{x},G_{\boldsymbol{y}}(\boldsymbol{x}))},\theta_{G_{\boldsymbol{z}}(\boldsymbol{y})},\theta_{G_{\boldsymbol{x}}(\boldsymbol{y},G_{\boldsymbol{z}}(\boldsymbol{y}))},\theta_D$     $\triangleright$ Initialize AMM parameters
**while** not done **do**
  $\boldsymbol{x}^{(1)},...,\boldsymbol{x}^{(M)}\sim q(\boldsymbol{x})$       $\triangleright$ Sample from data and priors
  $\boldsymbol{y}^{(1)},...,\boldsymbol{y}^{(M)}\sim p(\boldsymbol{y})$
  $\boldsymbol{z}^{(j)}\sim p(\boldsymbol{z}\,|\,\boldsymbol{y}=\boldsymbol{y}^{(j)}),\quad j=1,...,M$
  $\tilde{\boldsymbol{x}}^{(j)}\sim p(\boldsymbol{x}\,|\,\boldsymbol{y}=\boldsymbol{y}^{(j)},\boldsymbol{z}=\boldsymbol{z}^{(j)}),\quad j=1,...,M$    $\triangleright$ Sample from conditionals
  $\tilde{\boldsymbol{y}}^{(i)}\sim q(\boldsymbol{y}\,|\,\boldsymbol{x}=\boldsymbol{x}^{(i)}),\quad i=1,...,M$
  $\tilde{\boldsymbol{z}}^{(i)}\sim q(\boldsymbol{z}\,|\,\boldsymbol{x}=\boldsymbol{x}^{(i)},\boldsymbol{y}=\tilde{\boldsymbol{y}}^{(i)}),\quad i=1,...,M$
  $\rho_q^{(i)}\leftarrow D(\boldsymbol{x}^{(i)},\tilde{\boldsymbol{y}}^{(i)},\tilde{\boldsymbol{z}}^{(i)}),\quad i=1,...,M$    $\triangleright$ Compute discriminator predictions
  $\rho_p^{(j)}\leftarrow D(\tilde{\boldsymbol{x}}^{(j)},\boldsymbol{y}^{(j)},\boldsymbol{z}^{(j)}),\quad j=1,...,M$
  $\mathcal{L}_D\leftarrow -\frac{1}{M}\sum_{i=1}^M\log(\rho_q^{(i)})-\frac{1}{M}\sum_{j=1}^M log(1-\rho_p^{(j)})$   $\triangleright$ Compute discriminator losses
  $\mathcal{L}_{G_{\boldsymbol{x}}(\boldsymbol{y},G_{\boldsymbol{z}}(\boldsymbol{y}))}=\mathcal{L}_{G_{\boldsymbol{z}}(\boldsymbol{y})}\leftarrow -\frac{1}{M}\sum_{i=1}^M\log(\rho_p^{(i)})$   $\triangleright$ Compute x generator losses
  $\mathcal{L}_{G_{\boldsymbol{y}}(\boldsymbol{x})}=\mathcal{L}_{G_{\boldsymbol{z}}(\boldsymbol{x},G_{\boldsymbol{y}}(\boldsymbol{x}))}\leftarrow -\frac{1}{M}\sum_{i=1}^M\log(1-\rho_q^{(i)})$   $\triangleright$ Compute y and z generator loss
  $\theta_D\leftarrow\theta_D-\nabla_{\theta_D}\mathcal{L}_D$      $\triangleright$ Update discriminator parameters
  $\theta_{G_{\boldsymbol{x}}(\boldsymbol{y},G_{\boldsymbol{z}}(\boldsymbol{y}))}\leftarrow\theta_{G_{\boldsymbol{x}}(\boldsymbol{y},G_{\boldsymbol{z}}(\boldsymbol{y}))}-\nabla_{\theta_{G_{\boldsymbol{x}}(\boldsymbol{y},G_{\boldsymbol{z}}(\boldsymbol{y}))}}\mathcal{L}_{G_{\boldsymbol{x}}(\boldsymbol{y},G_{\boldsymbol{z}}(\boldsymbol{y}))}$   $\triangleright$ Update generator parameters
  $\theta_{G_{\boldsymbol{z}}(\boldsymbol{y})}\leftarrow\theta_{G_{\boldsymbol{z}}(\boldsymbol{y})}-\nabla_{\theta_{G_{\boldsymbol{z}}(\boldsymbol{y})}}\mathcal{L}_{G_{\boldsymbol{z}}(\boldsymbol{y})}$
  $\theta_{G_{\boldsymbol{y}}(\boldsymbol{x})}\leftarrow\theta_{G_{\boldsymbol{y}}(\boldsymbol{x})}-\nabla_{\theta_{G_{\boldsymbol{y}}(\boldsymbol{x})}}\mathcal{L}_{G_{\boldsymbol{y}}(\boldsymbol{x})}$
  $\theta_{G_{\boldsymbol{z}}(\boldsymbol{x},G_{\boldsymbol{y}}(\boldsymbol{x}))}\leftarrow\theta_{G_{\boldsymbol{z}}(\boldsymbol{x},G_{\boldsymbol{y}}(\boldsymbol{x}))}-\nabla_{\theta_{G_{\boldsymbol{z}}(\boldsymbol{x},G_{\boldsymbol{y}}(\boldsymbol{x}))}}\mathcal{L}_{G_{\boldsymbol{z}}(\boldsymbol{x},G_{\boldsymbol{y}}(\boldsymbol{x}))}$

---

### 2.3 SEMI-SUPERVISED ADVERSARIALLY LEARNED MIXTURE MODEL

The Semi-Supervised Adversarially Learned Mixture Model (SAMM) is an adversarial generative model for supervised or semi-supervised clustering and classification of data. The objective for training SAMM involves two adversarial games to match pairs of joint distributions. The supervised game matches inference distribution (4) to synthesis distribution (11) and is described by the following value function:

$$\min_G\max_D V(D,G)=\mathbb{E}_{q(\boldsymbol{x},\boldsymbol{y})}[\log(D(\boldsymbol{x},\boldsymbol{y},G_{\boldsymbol{z}}(\boldsymbol{x},\boldsymbol{y})))]+\mathbb{E}_{p(\boldsymbol{y},\boldsymbol{z})}[\log(1-D(G_{\boldsymbol{x}}(\boldsymbol{y},G_{\boldsymbol{z}}(\boldsymbol{y})),\boldsymbol{y},G_{\boldsymbol{z}}(\boldsymbol{y})))]$$

$$=\iiint q(\boldsymbol{x},\boldsymbol{y})q(\boldsymbol{z}\,|\,\boldsymbol{x},\boldsymbol{y})\log(D(\boldsymbol{x},\boldsymbol{y},\boldsymbol{z}))d\boldsymbol{x}d\boldsymbol{y}d\boldsymbol{z}$$

$$+\iiint p(\boldsymbol{y})p(\boldsymbol{z}\,|\,\boldsymbol{y})p(\boldsymbol{x}\,|\,\boldsymbol{y},\boldsymbol{z})\log(1-D(\boldsymbol{x},\boldsymbol{y},\boldsymbol{z}))d\boldsymbol{x}d\boldsymbol{y}d\boldsymbol{z}.$$

$$(14)$$

The unsupervised game matches either of the inference distributions, (6) or (7) to the synthesis distribution (11). In the case using distribution (6), the unsupervised game is described by (13). The generator for semi-supervised learning has three components: encoders $G_{\boldsymbol{z}}(\boldsymbol{x},G_{\boldsymbol{y}}(\boldsymbol{x}))$ and $G_{\boldsymbol{z}}(\boldsymbol{x},\boldsymbol{y})$ map the labeled and unlabeled data samples, respectively, to the latent space, and a decoder $G_{\boldsymbol{x}}(\boldsymbol{y},G_{\boldsymbol{z}}(\boldsymbol{y}))$ maps samples of $\boldsymbol{y}$ and $\boldsymbol{z}$ to the input space, where $G_{\boldsymbol{z}}(\boldsymbol{z})$ can either be a learned function or be specified by a prior. The encoder for labeled data again consists of two generators (Figure 1). A detailed description of the training algorithm is given in algorithm 2 of the appendix. In practice, optimization of each of the generators and the discriminator can be done simultaneously for both the unsupervised and semi-supervised updates.

## 3 RELATED WORKS

Unsupervised clustering using hybrid adversarial approaches are proposed by both Makhzani et al. (2016) (AAE) and Chen et al. (2016) (InfoGAN). For AAE, the synthesis generator is optimized

by minimizing the per-example L2 loss between between input data $\{\boldsymbol{x}_i\}$ and their reconstructions $\{\dot{\boldsymbol{x}}_i = G_{\boldsymbol{x}_i}(G_{\boldsymbol{y}}(\boldsymbol{x}_i), G_{\boldsymbol{z}}(\boldsymbol{x}_i))\}$, while the inference generator is optimized using both the L2 objective and an adversarial objective. For InfoGAN, the inference generator is optimized by maximizing the per-example Mutual Information (MI) between samples of categorical latent variables $\{\boldsymbol{y}_i \sim p(\boldsymbol{y})\}$ and continuous latent variables $\{\boldsymbol{z}_i \sim p(\boldsymbol{z})\}$ and their "reconstructions" $\{\{\dot{\boldsymbol{y}}_i, \dot{\boldsymbol{z}}_i\} = G_{\boldsymbol{y},\boldsymbol{z}}(G_{\boldsymbol{x}}(\boldsymbol{y}_i, \boldsymbol{z}_i))\}$, while the synthesis generator is optimized using both the MI objective and an adversarial objective.

Recent approaches using self-supervision with data augmentation have also been proposed for unsupervised discrete representation learning by Kilinc & Uysal (2018) (LALNets) and Hu et al. (2017) (IMSAT). In IMSAT, a network is trained to maximize the Mutual Information between input data and a discrete representation, similar to InfoGAN. In addition to this objective, the network is regularized by encouraging the representation of original, unperturbed data $\{\boldsymbol{x}_i\}$ to be close to that of transformed data $\{T(\boldsymbol{x}_i)\}$. In the LALNets framework, a network is trained to distinguish unperturbed data and augmented data, and then k-means clustering is performed on the latent space representation of the unperturbed data to obtain cluster labels.

On the other end of the generative spectrum, Dilokthanakul et al. (2016) and Jiang et al. (2017) offer non-adversarial, VAE-based approaches for unsupervised clustering. Like in the AMM, the combination of priors for the latent variables $\boldsymbol{y}$ and $\boldsymbol{z}$ is modeled as a Gaussian mixture model, where $\boldsymbol{y}$ corresponds to the mixture components.

Multiple adversarial methodologies have been proposed for supervised or semi-supervised learning (Springenberg, 2015; Salimans et al., 2016; Miyato et al., 2017; Dai et al., 2017), but they suffer from the same limitation as the original GAN: they do not provide inference. Gan et al. (2017), Li et al. (2017) and Deng et al. (2017) introduce a third player to the adversarial game. Although this extra player allows to infer categorical variables, these approaches are not fully adversarial as auxiliary "collaborative" terms are added to the objective function. Moreover, categorical and continuous latent variables are modeled independently.

The adversarial and hybrid-adversarial approaches thus far discussed all model $\boldsymbol{y}$ and $\boldsymbol{z}$ as being conditionally independent from each other. This may be an ideal prior structure for inference, for example, in learning disentangled representations of $\boldsymbol{x}$ sampled from a limited domain (Chen et al., 2016). However, the independence assumption cannot account for the notion of proximity between categories because $\boldsymbol{z}$ is identically distributed for each category in $\boldsymbol{y}$. Therefore, the distance between categories is equal and indeterminate. AMM and SAMM are presented as adversarial approaches to model conditional dependencies between $\boldsymbol{y}$ and $\boldsymbol{z}$, but they do not preclude the independence assumption. The proposed methods can model $\boldsymbol{y}$ and $\boldsymbol{z}$ as conditionally independent with inference distribution

$$q(\boldsymbol{x}, \boldsymbol{y}, \boldsymbol{z}) = q(\boldsymbol{x})q(\boldsymbol{y} \,|\, \boldsymbol{x})q(\boldsymbol{z} \,|\, \boldsymbol{x}), \tag{15}$$

and synthesis distribution

$$p(\boldsymbol{x}, \boldsymbol{y}, \boldsymbol{z}) = p(\boldsymbol{y})p(\boldsymbol{z})p(\boldsymbol{x} \,|\, \boldsymbol{y}, \boldsymbol{z}); \tag{16}$$

however, analysis of this graphical model is left for future work.

## 4 EVALUATION

AMM and SAMM are evaluated using two image datasets: MNIST (LeCun & Cortes, 2010) and SVHN (Netzer et al., 2011). The provided training and testing splits are used for MNIST experiments with 5000 randomly selected examples left out of the training set for validation. The same training, testing, and validation splits as Dumoulin et al. (2016) are used for SVHN. Preprocessing is limited to scaling image intensities on the range $[0, 1]$. Detailed architectures for each experiment are shown in Section B of the appendix. We optimize all networks using Adam (Kingma & Ba, 2014) with $\alpha = 0.0002$ and $\beta_1 = 0.5$. All kernel weights are initialized using a Gaussian distribution with standard deviation 0.02, all biases are initialized to 0.0. Following the criteria in Jiang et al. (2017), the performance is evaluated as follows:

$$ACC = \max_{n \in N} \frac{\sum_{i=1}^{M} \mathbb{1}\{y_i = n(\tilde{y}_{\tilde{z},i})\}}{M}, \tag{17}$$

where $M$ is the number of samples, $y_i$ is the true label, and $\tilde{y}_{\tilde{z},i}$ is the predicted cluster label, and $N$ is the set of all mappings between cluster labels and true labels.

### 4.1 GRADIENT PENALTY

The gradient penalty introduced by Gulrajani et al. (2017) is added to the discriminator loss to help stabilize training of AMM and SAMM models. This penalty keeps the gradients of the discriminator with respect to the inputs $x$, $y$, and $z$ on the same order of magnitude. The penalty applied to the discriminator loss is

$$\mathcal{L}_{\nabla_{\hat{x},\hat{y},\hat{z}}} = \lambda \underset{(\hat{x},\hat{y},\hat{z})\sim\mathbb{P}_{\hat{x},\hat{y},\hat{z}}}{\mathbb{E}} \left[ (||\nabla_{\hat{x},\hat{y},\hat{z}} D(\hat{x},\hat{y},\hat{z})||_2 - 1)^2 \right], \tag{18}$$

where points $(\hat{x},\hat{y},\hat{z})$ are drawn at random on straight lines between real or prior samples $(x,\tilde{y},\tilde{z})$ and synthesized or inferred samples $(\tilde{x},y,z)$. The gradient penalty for Jensen-Shannon GAN introduced by Roth et al. (2017) has also been explored, but did not produce better results. The regularization term is set to $\lambda = 10.0$, and $\lambda = 0.01$ for MNIST and SVHN experiments, respectively.

### 4.2 MNIST

In this section, the AMM is evaluated on the task of unsupervised clustering of hand-drawn digits using the MNIST dataset. To model $p(y)p(z\,|\,y)$, a 64 dimensional mixture of Gaussians is used with 10, 20, and 30-components across 3 experiments. A multinomial prior is used for $p(y)$ with uniform probability for each class. The mean and variance of each component distribution are learned using the reparameterization trick via (12). Table 1 reports the test-set clustering error-rate mean and standard deviation over 10 trials, with the 10-component AMM achieving a $3.32 \pm 0.39$ percent error rate. Figure 2 visualizes results from 1 of the 10 trials.

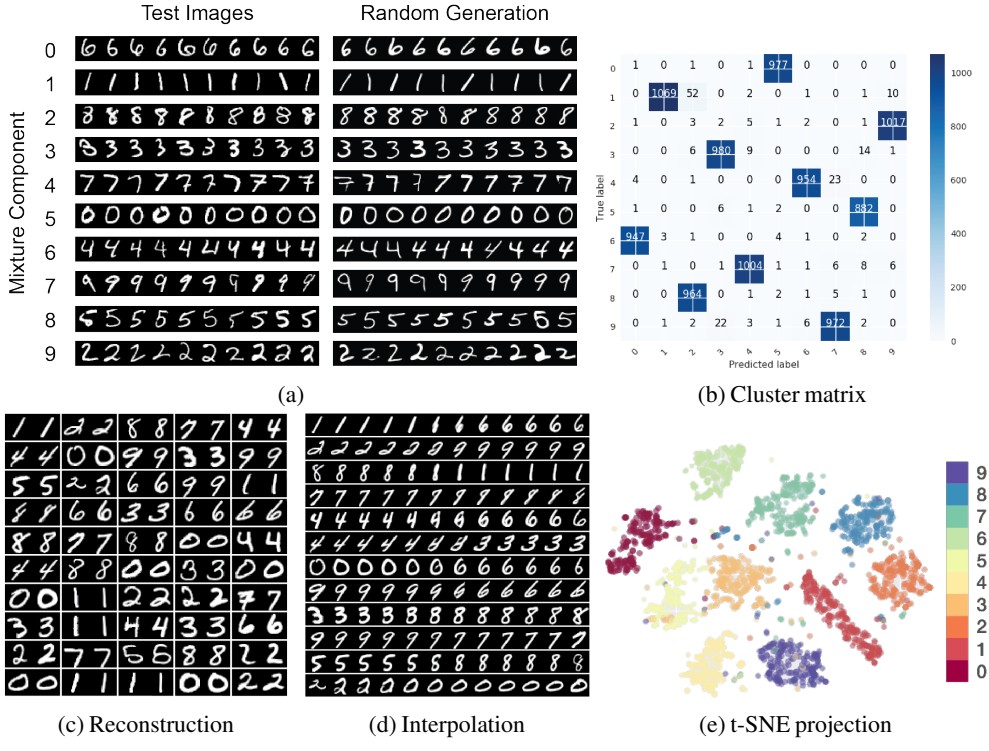

Figure 2: Unsupervised clustering of MNIST data with 10 mixture components. *(a)* Comparing test image membership and randomly generated digits for each mixture component. *(b)* Cluster matrix. *(c)* Reconstructions of input images: original data on the left of each pair. *(d)* Interpolation between examples: original data samples are shown in the first and last columns with linearly interpolated generations between. *(e)* t-SNE projection of testing samples, color-coded for the MNIST class labels (0 to 9).

Table 1: Test set clustering error rate and standard deviation for MNIST data. Methods using data augmentation are denoted with $*$.

| MODEL | K | MNIST | SVHN |
|---|---|---|---|
| **GMVAE** (DILOKTHANAKUL ET AL., 2016) | 30 | **7.23**±**1.60** | - |
| **VADE** (JIANG ET AL., 2017) | 10 | **5.54** | - |
| **CATGAN** (SPRINGENBERG, 2015) | 20 | **9.70** | - |
| **INFOGAN** (CHEN ET AL., 2016) | 10 | **5.00** | - |
| **AAE** (MAKHZANI ET AL., 2016) | 30 | 4.10±1.13 | - |
| **IMSAT** (HU ET AL., 2017) $*$ | 10 | 1.59±0.40 | 42.7±3.90 |
| **LALNETS** (KILINC & UYSAL, 2018) $*$ | 10 | 1.68±0.08 | 23.2±1.30 |
| **AMM** | 10 | 3.32±0.39 | - |
| **AMM** | 20 | 3.99±0.79 | 35.0±5.24 |
| **AMM** | 30 | 4.01±1.11 | 20.4±0.80 |

## 4.3 SVHN

### 4.3.1 UNSUPERVISED CLUSTERING

In this section, unsupervised clustering is revisited. The SVHN dataset is used to investigate how confounding attributes, such as color and contrast, affects the semantic separation of digits. To model $p(\boldsymbol{y})p(\boldsymbol{z}\,|\,\boldsymbol{y})$ a 64 dimensional mixture of Gaussians is used with 10, 20, and 30-components across 3 experiments. A multinomial prior is used for $p(\boldsymbol{y})$ with uniform probability for each class, and the mean and variance of each component distribution are learned using the reparameterization trick via (12).

Figure 3a shows random samples drawn from each component distribution generated by $G_{\boldsymbol{x}}$. Figure 3b is a t-SNE projection of test samples drawn from $G_{\boldsymbol{z}}$ onto a 2D manifold. We show in 3c that AMM learns a smooth latent manifold as we interpolate between examples from SVHN. The cluster matrix for the SVHN test set is shown in figure 3d, demonstrating an overall classification error rate of $20.4\%\pm0.80$. From these results, we can make a few observations. First, the generated images are realistic looking. This is the case for every cluster other than cluster 5, which appears to be a noise cluster. Secondly, each cluster only contains a single digit. Finally, we observe that for each digit there are multiple clusters, one containing a dark background with light numbers (ie. cluster 28), and another containing a light background with dark numbers (ie. cluster 12). This is important, as it shows that the AMM has learned semantically meaningful clusters.

### 4.3.2 SEMI-SUPERVISED CLUSTERING AND CLASSIFICATION

It is evident from the last experiment that the confounders introduced by the SVHN dataset made unsupervised semantic clustering more difficult. In this section we show how SAMM can be used to guide clustering along predefined categories using only a small amount of labeled data. To this end, we limit the samples drawn from $q(\boldsymbol{x},\boldsymbol{y})$ to a random selection of 1000 examples from the training set. To model $p(\boldsymbol{y})p(\boldsymbol{z}\,|\,\boldsymbol{y})$ we use a 64 dimensional mixture of 10 spherical Gaussians, where the the mean and variance of each component distribution are learned using the reparameterization trick via (12). There is considerable class imbalance in the SVHN dataset, so a multinomial prior is used for $p(\boldsymbol{y})$ with each class probability set to the frequency observed in the labeled subset of the training data.

Table 2 reports the test-set error-rate mean and variance over 10 trials. SAMM achieves $5.60\pm0.45$ percent error rate, which is an improvement over the ALI baseline. Figure 4 shows visualizations of results from 1 of the 10 trials. Finally, given that we have defined $p(\boldsymbol{y})p(\boldsymbol{z}\,|\,\boldsymbol{y})$ we can use Bayes' theorem to derive $p(\boldsymbol{y}\,|\,\boldsymbol{z})$ and get a classifier given an image embedding $\tilde{\boldsymbol{z}}$:

$$\tilde{\boldsymbol{y}}_{\tilde{\boldsymbol{z}}} = \operatorname*{argmax}_{k}[p(\boldsymbol{z}=\tilde{\boldsymbol{z}}\,|\,\boldsymbol{y}=k)p(\boldsymbol{y}=k)] \tag{19}$$

Figures 4e and 4f compare the confusion matrices for predictions given by $\tilde{\boldsymbol{y}}_{\tilde{\boldsymbol{z}}}$ and those given by $\tilde{\boldsymbol{y}}$ from $G_{\boldsymbol{y}}$. The similarity between each is further evidence that the inference network has learned to embed data according to the desired distribution.

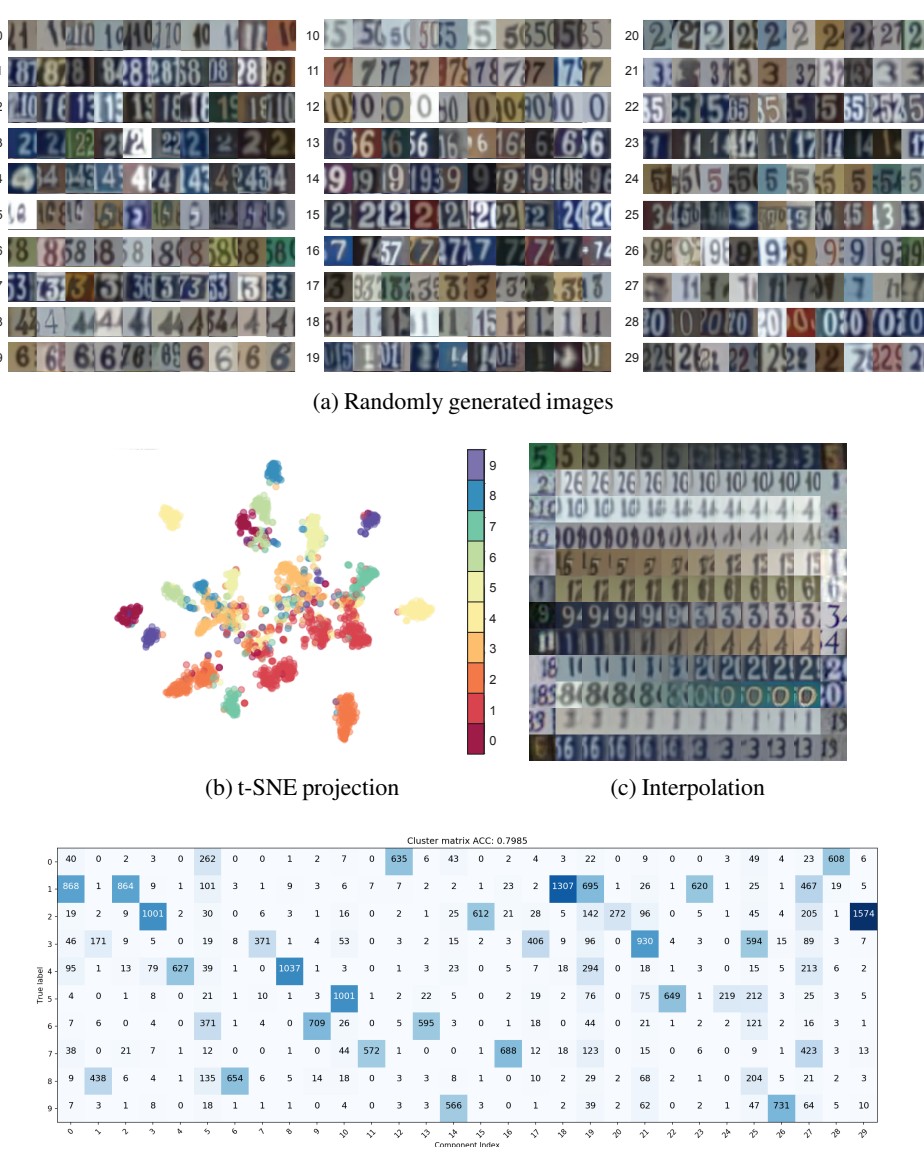

Figure 3: Unsupervised clustering of SVHN data with 30 mixture components. *(a)* Randomly generated images for each mixture component, mixture component indices are indicated to the left of each row. *(b)* t-SNE projection of testing samples, color-coded for the SVHN class label (0 to 9). *(c)* Interpolation between examples: input images are shown in the first and last columns. *(d)* Cluster matrix.

## 5 CONCLUSION

The AMM is presented as a generative model for unsupervised or semi-supervised data clustering with several contributions. The AMM is the first fully adversarially optimized method to model the conditional dependence between categorical and continuous latent variables, providing impressive unsupervised clustering and competitive semi-supervised classification results on benchmark datasets. In contrast with other semi-supervised approaches, we have shown that the use of additional losses or discriminators (Gan et al. (2017); Li et al. (2017); Deng et al. (2017)) are redundant additions to frameworks, as the AMM yields similar or better results to these methods. In contrast with Dai et al. (2017), our strong semi-supervised performance and qualitatively good image generation demonstrate that semi-supervised performance and image generation are not necessarily opposing goals. As a fully adversarial framework, the AMM provides a simple, yet powerful formulation as a foundation for

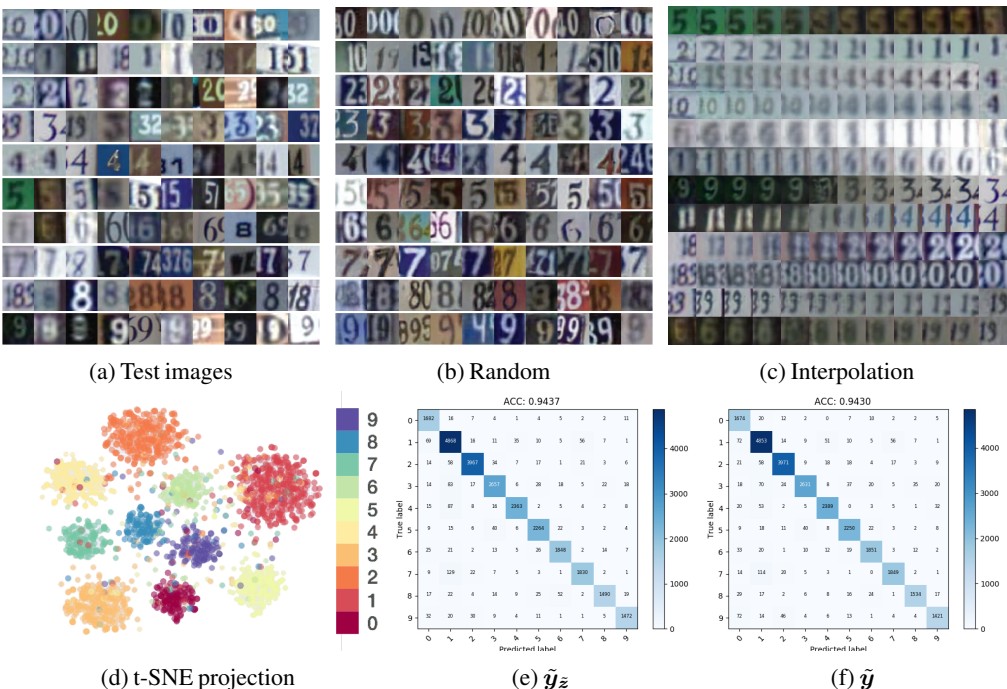

Figure 4: Semi-supervised clustering and classification of SVHN data with 10 mixture components. *(a)* Test image predictions: rows correspond to the predicted class. *(b)* Randomly generated images for each mixture component. *(c)* Interpolation between examples: original data samples in first and last columns. *(d)* t-SNE projection of testing samples, color-coded for the SVHN class label (0 to 9). Confusion matrix for predictions given an image embedding *(e)* and given the generator $G_{\boldsymbol{y}}$ *(e)*.

Table 2: Semi-supervised test set missclassification rate and standard deviation for SVHN data. † and ‡ denote similar encoder/classifier architectures.

| MODEL | SVHN ($N = 1000$) |
|---|---|
| **AAE** (MAKHZANI ET AL., 2016) | **17.70±0.24** |
| **IMPROVEDGAN** (SALIMANS ET AL., 2016) ] † | **8.11±1.30** |
| **ALI** (DUMOULIN ET AL., 2016) | **7.42±0.65** |
| **VAT SMALL** (MIYATO ET AL., 2017) † | **6.83±0.24** |
| **TRIPLEGAN** (LI ET AL., 2017) ‡ | **5.77±0.17** |
| **SGAN** (DENG ET AL., 2017) ‡ | **5.73±0.12** |
| **VAT LARGE** (MIYATO ET AL., 2017) ‡ | **4.28±0.10** |
| (DAI ET AL., 2017) † | **4.25±0.03** |
| **SAMM** † | **5.60±0.45** |

future work. For example, the distribution of mixture components can be something be other than a Gaussian without the need to derive an new evidence lower bound. Furthermore, by learning conditional dependencies between $\boldsymbol{y}$ and $\boldsymbol{z}$, the AMM preserves the notion of proximity between classes in the latent space, which could be useful for applications in metric space learning and few shot learning. On top of this, learning the mean and variance of mixture components gives access to subsequent analysis of the likelihood function, which could provide an interpretable insight of the learned distributions. With that, the mixture coefficients are the only terms to be integrated into the framework. As a whole, the AMMs demonstrably strong performance validates its use in future work and extension into other domains.

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

## APPENDIX A    SAMM ALGORITHM

Algorithm 2 outlines the SAMM training procedure.

---

**Algorithm 2** SAMM training procedure using distributions (4), (6), and (11).

---

$\theta_{G_{\boldsymbol{y}}(\boldsymbol{x})}, \theta_{G_{\boldsymbol{z}}(\boldsymbol{x}, G_{\boldsymbol{y}}(\boldsymbol{x}))}, \theta_{G_{\boldsymbol{z}}(\boldsymbol{y})}, \theta_{G_{\boldsymbol{x}}(\boldsymbol{y}, G_{\boldsymbol{z}}(\boldsymbol{y}))}, \theta_D$      ▷ Initialize SAMM parameters

**while** not done **do**

$\quad \boldsymbol{x}_u^{(1)}, ..., \boldsymbol{x}_u^{(M)} \sim q(\boldsymbol{x})$      ▷ Sample from unlabeled data and priors

$\quad \boldsymbol{y}_u^{(1)}, ..., \boldsymbol{y}_u^{(M)} \sim p(\boldsymbol{y})$

$\quad \boldsymbol{z}_u^{(j)} \sim p(\boldsymbol{z} \,|\, \boldsymbol{y} = \boldsymbol{y}_u^{(j)}), \quad j = 1, ..., M$

$\quad \tilde{\boldsymbol{x}}_u^{(j)} \sim p(\boldsymbol{x} \,|\, \boldsymbol{y} = \boldsymbol{y}_u^{(j)}, \boldsymbol{z} = \boldsymbol{z}_u^{(j)}), \quad j = 1, ..., M$      ▷ Sample from conditionals

$\quad \tilde{\boldsymbol{y}}_u^{(i)} \sim q(\boldsymbol{y} \,|\, \boldsymbol{x} = \boldsymbol{x}_u^{(i)}), \quad i = 1, ..., M$

$\quad \tilde{\boldsymbol{z}}_u^{(i)} \sim q(\boldsymbol{z} \,|\, \boldsymbol{x} = \boldsymbol{x}_u^{(i)}, \boldsymbol{y} = \tilde{\boldsymbol{y}}_u^{(i)}), \quad i = 1, ..., M$

$\quad \left( \boldsymbol{x}_\ell^{(1)}, ..., \boldsymbol{x}_\ell^{(M)} \right), \left( \tilde{\boldsymbol{y}}_\ell^{(1)}, ..., \tilde{\boldsymbol{y}}_\ell^{(M)} \right) \sim q(\boldsymbol{x}, \boldsymbol{y})$      ▷ Sample from labeled data and priors

$\quad \boldsymbol{y}_\ell^{(1)}, ..., \boldsymbol{y}_\ell^{(M)} \sim p(\boldsymbol{y})$

$\quad \boldsymbol{z}_\ell^{(j)} \sim p(\boldsymbol{z} \,|\, \boldsymbol{y} = \boldsymbol{y}_\ell^{(j)}), \quad j = 1, ..., M$

$\quad \tilde{\boldsymbol{x}}_\ell^{(j)} \sim p(\boldsymbol{x} \,|\, \boldsymbol{y} = \boldsymbol{y}_\ell^{(j)}, \boldsymbol{z} = \boldsymbol{z}_\ell^{(j)}), \quad j = 1, ..., M$      ▷ Sample from conditionals

$\quad \tilde{\boldsymbol{z}}_\ell^{(i)} \sim q(\boldsymbol{z} \,|\, \boldsymbol{x} = \boldsymbol{x}_\ell^{(i)}, \boldsymbol{y} = \tilde{\boldsymbol{y}}_\ell^{(i)}), \quad i = 1, ..., M$

$\quad \boldsymbol{\rho}_{q_u}^{(i)} \leftarrow D(\boldsymbol{x}_u^{(i)}, \tilde{\boldsymbol{y}}_u^{(i)}, \tilde{\boldsymbol{z}}_u^{(i)}), \quad i = 1, ..., M$      ▷ Compute predictions for unlabeled data

$\quad \boldsymbol{\rho}_{p_u}^{(j)} \leftarrow D(\tilde{\boldsymbol{x}}_u^{(j)}, \boldsymbol{y}_u^{(j)}, \boldsymbol{z}_u^{(j)}), \quad j = 1, ..., M$

$\quad \boldsymbol{\rho}_{q_\ell}^{(i)} \leftarrow D(\boldsymbol{x}_\ell^{(i)}, \tilde{\boldsymbol{y}}_\ell^{(i)}, \tilde{\boldsymbol{z}}_\ell^{(i)}), \quad i = 1, ..., M$      ▷ Compute predictions for labeled data

$\quad \boldsymbol{\rho}_{p_\ell}^{(j)} \leftarrow D(\tilde{\boldsymbol{x}}_\ell^{(j)}, \boldsymbol{y}_\ell^{(j)}, \boldsymbol{z}_\ell^{(j)}), \quad j = 1, ..., M$

$\quad \mathcal{L}_{D_u} \leftarrow -\frac{1}{2M} \sum_{i=1}^M \log(\boldsymbol{\rho}_{q_u}^{(i)}) - \frac{1}{2M} \sum_{j=1}^M \log(1 - \boldsymbol{\rho}_{p_u}^{(j)})$      ▷ Compute discriminator losses

$\quad \mathcal{L}_{D_\ell} \leftarrow -\frac{1}{2M} \sum_{i=1}^M \log(\boldsymbol{\rho}_{q_\ell}^{(i)}) - \frac{1}{2M} \sum_{j=1}^M \log(1 - \boldsymbol{\rho}_{p_\ell}^{(j)})$

$\quad \mathcal{L}_{G_{\boldsymbol{y}_u}(\boldsymbol{x})} = \mathcal{L}_{G_{\boldsymbol{z}_u}(\boldsymbol{x}, G_{\boldsymbol{y}}(\boldsymbol{x}))} \leftarrow -\frac{1}{2M} \sum_{i=1}^M \log(1 - \boldsymbol{\rho}_{q_u}^{(i)})$      ▷ Compute inference losses

$\quad \mathcal{L}_{G_{\boldsymbol{z}_\ell}(\boldsymbol{x}, G_{\boldsymbol{y}}(\boldsymbol{x}))} \leftarrow -\frac{1}{2M} \sum_{i=1}^M \log(1 - \boldsymbol{\rho}_{q_\ell}^{(i)})$

$\quad \mathcal{L}_{G_{\boldsymbol{x}_u}(\boldsymbol{y}, G_{\boldsymbol{z}}(\boldsymbol{y}))} = \mathcal{L}_{G_{\boldsymbol{z}_u}(\boldsymbol{y})} \leftarrow -\frac{1}{2M} \sum_{i=1}^M \log(\boldsymbol{\rho}_{p_u}^{(i)})$      ▷ Compute $\boldsymbol{x}$ generator losses

$\quad \mathcal{L}_{G_{\boldsymbol{x}_\ell}(\boldsymbol{y}, G_{\boldsymbol{z}}(\boldsymbol{y}))} = \mathcal{L}_{G_{\boldsymbol{z}_\ell}(\boldsymbol{y})} \leftarrow -\frac{1}{2M} \sum_{i=1}^M \log(\boldsymbol{\rho}_{p_\ell}^{(i)})$

$\quad \theta_D \leftarrow \theta_D - \nabla_{\theta_D}(\mathcal{L}_{D_u} + \mathcal{L}_{D_\ell})$      ▷ Update discriminator parameters

$\quad \theta_{G_{\boldsymbol{z}}(\boldsymbol{x}, G_{\boldsymbol{y}}(\boldsymbol{x}))} \leftarrow \theta_{G_{\boldsymbol{z}}(\boldsymbol{x}, G_{\boldsymbol{y}}(\boldsymbol{x}))} - \nabla_{\theta_{G_{\boldsymbol{z}}(\boldsymbol{x}, G_{\boldsymbol{y}}(\boldsymbol{x}))}} \left( \mathcal{L}_{G_{\boldsymbol{z}_u}(\boldsymbol{x}, G_{\boldsymbol{y}}(\boldsymbol{x}))} + \mathcal{L}_{G_{\boldsymbol{z}_\ell}(\boldsymbol{x}, G_{\boldsymbol{y}}(\boldsymbol{x}))} \right)$      ▷
Update $\boldsymbol{z}$ inference parameters

$\quad \theta_{G_{\boldsymbol{y}}(\boldsymbol{x})} \leftarrow \theta_{G_{\boldsymbol{y}}(\boldsymbol{x})} - \nabla_{\theta_{G_{\boldsymbol{y}}(\boldsymbol{x})}} \mathcal{L}_{G_{\boldsymbol{y}_u}(\boldsymbol{x})}$      ▷ Update $\boldsymbol{y}$ inference parameters

$\quad \theta_{G_{\boldsymbol{x}}(\boldsymbol{y}, G_{\boldsymbol{z}}(\boldsymbol{y}))} \leftarrow \theta_{G_{\boldsymbol{x}}(\boldsymbol{y}, G_{\boldsymbol{z}}(\boldsymbol{y}))} - \nabla_{\theta_{G_{\boldsymbol{x}}(\boldsymbol{y}, G_{\boldsymbol{z}}(\boldsymbol{y}))}} \left( \mathcal{L}_{G_{\boldsymbol{x}_u}(\boldsymbol{y}, G_{\boldsymbol{z}}(\boldsymbol{y}))} + \mathcal{L}_{G_{\boldsymbol{x}_\ell}(\boldsymbol{y}, G_{\boldsymbol{z}}(\boldsymbol{y}))} \right)$      ▷
Update $\boldsymbol{x}$ synthesis parameters

$\quad \theta_{G_{\boldsymbol{z}}(\boldsymbol{y})} \leftarrow \theta_{G_{\boldsymbol{z}}(\boldsymbol{y})} - \nabla_{\theta_{G_{\boldsymbol{z}}(\boldsymbol{y})}} \left( \mathcal{L}_{G_{\boldsymbol{z}_u}(\boldsymbol{y})} + \mathcal{L}_{G_{\boldsymbol{z}_\ell}(\boldsymbol{y})} \right)$      ▷ Update $\boldsymbol{z}$ synthesis parameters

---

## APPENDIX B    EXPERIMENT INFORMATION

### B.1    MODEL ARCHITECTURES

Tables 3, 4, 5 and 6 detail the model architectures for the MNIST experiments. Tables 7, 8, 9 and 10 detail the model architectures for the SVHN experiments. Model inputs and outputs are highlighted in boldface. "Leak" denotes Leaky ReLU activations. "ExpN" denotes the following activation function:

$$\text{ExpN}(\boldsymbol{x}) = \exp(0.5\,\boldsymbol{x}) \odot \epsilon, \quad \epsilon \sim \mathcal{N}(0, I). \tag{20}$$

Table 3: MNIST: $G_{\boldsymbol{z}}(\boldsymbol{x})G_{\boldsymbol{y}}(\boldsymbol{x},G_{\boldsymbol{z}}(\boldsymbol{x}))$

| Name | Input | Channels | Width | Stride | Dropout | BatchNorm | Activation |
|------|-------|----------|-------|--------|---------|-----------|------------|
| $\mathbf{x}$ | - | 1 | - | - | - | - | - |
| $y_1$ | $\mathbf{x}$ | 32 | 2 | 1 | 0.2 | - | - |
| $z_1$ | $\mathbf{x}$ | 32 | 2 | 1 | 0.2 | - | - |
| $y_{1a}$ | $y_1 + z_1$ | 32 | - | - | - | yes | Leak 0.2 |
| $z_{1a}$ | $z_1$ | 32 | - | - | - | yes | Leak 0.2 |
| $y_2$ | $y_{1a}$ | 32 | 3 | 2 | 0.2 | - | - |
| $z_2$ | $z_{1a}$ | 32 | 3 | 2 | 0.2 | - | - |
| $y_{2a}$ | $y_2 + z_2$ | 32 | - | - | - | yes | Leak 0.2 |
| $z_{2a}$ | $z_2$ | 32 | - | - | - | yes | Leak 0.2 |
| $y_3$ | $y_{2a}$ | 64 | 3 | 2 | 0.2 | - | - |
| $z_3$ | $z_{2a}$ | 64 | 3 | 2 | 0.2 | - | - |
| $y_{3a}$ | $y_3 + z_3$ | 64 | - | - | - | yes | Leak 0.2 |
| $z_{3a}$ | $z_3$ | 64 | - | - | - | yes | Leak 0.2 |
| $y_4$ | $y_{3a}$ | 64 | 3 | 1 | 0.2 | - | - |
| $z_4$ | $z_{3a}$ | 64 | 3 | 1 | 0.2 | - | - |
| $y_{4a}$ | $y_4 + z_4$ | 64 | - | - | - | yes | Leak 0.2 |
| $z_{4a}$ | $z_4$ | 64 | - | - | - | yes | Leak 0.2 |
| $y_5$ | $y_{4a}$ | 128 | 4 | 1 | 0.2 | - | - |
| $z_5$ | $z_{4a}$ | 128 | 4 | 1 | 0.2 | - | - |
| $y_{5a}$ | $y_5 + z_5$ | 128 | - | - | - | yes | Leak 0.2 |
| $z_{5a}$ | $z_5$ | 128 | - | - | - | yes | Leak 0.2 |
| $y_\mu$ | $y_{5a}$ | k | 1 | 1 | - | - | - |
| $y_\sigma$ | $y_{5a}$ | k | 1 | 1 | - | - | ExpN |
| $\mathbf{y}$ | $y_\mu + y_\sigma$ | k | - | - | - | - | Softmax |
| $z_\mu$ | $z_{5a}$ | 64 | 1 | 1 | - | - | - |
| $z_\sigma$ | $z_{5a}$ | 64 | 1 | 1 | - | - | ExpN |
| $\mathbf{z}$ | $z_\mu + z_\sigma$ | 64 | - | - | - | - | - |

Table 4: MNIST: $G_{\boldsymbol{z}}(\boldsymbol{y})$

| Name | Input | Channels | Width | Stride | Dropout | BatchNorm | Activation |
|------|-------|----------|-------|--------|---------|-----------|------------|
| $\mathbf{y}$ | - | k | - | - | - | - | - |
| $z_1$ | $\mathbf{y}$ | 64 | 1 | 1 | - | yes | Leak 0.2 |
| $z_2$ | $z_1$ | 64 | 1 | 1 | 0.2 | yes | Leak 0.2 |
| $z_\mu$ | $z_2$ | 64 | 1 | 1 | - | - | - |
| $z_\sigma$ | zeros | 64 | - | - | - | - | ExpN |
| $\mathbf{z}$ | $z_\mu + z_\sigma$ | 64 | - | - | - | - | - |

Table 5: MNIST: $G_{\boldsymbol{x}}(\boldsymbol{y}, G_{\boldsymbol{z}}(\boldsymbol{y}))$

| Name | Input | Channels | Width | Stride | Dropout | BatchNorm | Activation |
|------|-------|----------|-------|--------|---------|-----------|------------|
| $\mathbf{y}$ | - | k | - | - | - | - | - |
| $\mathbf{z}$ | - | 64 | - | - | - | - | - |
| $y_1$ | $\mathbf{y}$ | 128 | 1 | 1 | - | - | - |
| $z_1$ | $\mathbf{z}$ | 128 | 1 | 1 | - | - | - |
| $y_{1a}$ | $y_1 + z_1$ | 128 | - | - | - | yes | - |
| $z_{1a}$ | $z_1$ | 128 | - | - | - | yes | Leak 0.2 |
| $y_2$ | $y_{1a}$ | 64 | 4 | 1 | 0.2 | - | - |
| $z_2$ | $z_{1a}$ | 64 | 4 | 1 | 0.2 | - | - |
| $y_{2a}$ | $y_2 + z_2$ | 64 | - | - | - | yes | - |
| $z_{2a}$ | $z_2$ | 64 | - | - | - | yes | Leak 0.2 |
| $y_3$ | $y_{2a}$ | 64 | 3 | 1 | 0.2 | - | - |
| $z_3$ | $z_{2a}$ | 64 | 3 | 1 | 0.2 | - | - |
| $y_{3a}$ | $y_3 + z_3$ | 64 | - | - | - | yes | - |
| $z_{3a}$ | $z_3$ | 64 | - | - | - | yes | Leak 0.2 |
| $y_4$ | $y_{3a}$ | 32 | 3 | 2 | 0.2 | - | - |
| $z_4$ | $z_{3a}$ | 32 | 3 | 2 | 0.2 | - | - |
| $y_{4a}$ | $y_4 + z_4$ | 32 | - | - | - | yes | - |
| $z_{4a}$ | $z_4$ | 32 | - | - | - | yes | Leak 0.2 |
| $y_5$ | $y_{4a}$ | 32 | 3 | 2 | 0.2 | - | - |
| $z_5$ | $z_{4a}$ | 32 | 3 | 2 | 0.2 | - | - |
| $y_{5a}$ | $y_5 + z_5$ | 32 | - | - | - | yes | - |
| $z_{5a}$ | $z_5$ | 32 | - | - | - | yes | Leak 0.2 |
| $y_6$ | $y_{5a}$ | 1 | 2 | 1 | 0.2 | - | - |
| $z_6$ | $z_{5a}$ | 1 | 2 | 1 | 0.2 | - | - |
| $\mathbf{x}$ | $y_6 + z_6$ | 1 | - | - | - | - | Sigmoid |

Table 6: MNIST: $D(\boldsymbol{x}, \boldsymbol{y}, \boldsymbol{z})$

| Name | Input | Channels | Width | Stride | Dropout | BatchNorm | Activation |
|------|-------|----------|-------|--------|---------|-----------|------------|
| $\mathbf{x}$ | - | 1 | - | - | - | - | - |
| $\mathbf{y}$ | - | k | - | - | - | - | - |
| $\mathbf{z}$ | - | 64 | - | - | - | - | - |
| $x_1$ | $\mathbf{x}$ | 32 | 2 | 1 | - | - | Leak 0.2 |
| $x_2$ | $x_1$ | 32 | 3 | 2 | 0.2 | - | Leak 0.2 |
| $x_3$ | $x_2$ | 64 | 3 | 2 | 0.2 | - | Leak 0.2 |
| $x_4$ | $x_3$ | 64 | 3 | 1 | 0.2 | - | Leak 0.2 |
| $x_5$ | $x_4$ | 128 | 4 | 1 | 0.2 | - | Leak 0.2 |
| $y_1$ | $\mathbf{y}$ | 64 | 1 | 1 | - | - | Leak 0.2 |
| $y_2$ | $y_1$ | 64 | 1 | 1 | 0.2 | - | Leak 0.2 |
| $z_1$ | $\mathbf{z}$ | 64 | 1 | 1 | - | - | Leak 0.2 |
| $z_2$ | $z_1$ | 64 | 1 | 1 | 0.2 | - | Leak 0.2 |
| $p_0$ | $x_5 \mid y_2 \mid z_2$ | 256 | - | - | - | - | - |
| $p_1$ | $p_0$ | 256 | 1 | 1 | 0.2 | - | Leak 0.2 |
| $p_2$ | $p_1$ | 256 | 1 | 1 | 0.2 | - | Leak 0.2 |
| $\mathbf{p}$ | $p_1$ | 1 | 1 | 1 | 0.2 | - | - |

Table 7: SVHN: $G_z(x)G_y(x,G_z(x))$

| Name | Input | Channels | Width | Stride | Dropout | BatchNorm | Activation |
|------|-------|----------|-------|--------|---------|-----------|------------|
| **x** | - | 1 | - | - | - | - | - |
| $x_1$ | **x** | 96 | 3 | - | - | yes | Leak 0.2 |
| $x_2$ | $x_1$ | 96 | 3 | - | - | yes | Leak 0.2 |
| $x_3$ | $x_2$ | 96 | 3 | Max2 | - | yes | Leak 0.2 |
| $x_4$ | $x_3$ | 192 | 3 | - | - | yes | Leak 0.2 |
| $x_5$ | $x_4$ | 192 | 3 | - | - | yes | Leak 0.2 |
| $x_6$ | $x_5$ | 192 | 3 | Max2 | - | yes | Leak 0.2 |
| $x_7$ | $x_6$ | 384 | 3 | - | - | yes | Leak 0.2 |
| $y_1$ | $x_7$ | 192 | 1 | - | - | - | - |
| $z_1$ | $x_7$ | 192 | 1 | - | - | - | - |
| $y_{1a}$ | $y_1 + z_1$ | 192 | - | - | - | yes | Leak 0.2 |
| $z_{1a}$ | $z_1$ | 192 | - | - | - | yes | Leak 0.2 |
| $y_2$ | $y_{1a}$ | 96 | 1 | - | - | - | - |
| $z_2$ | $z_{1a}$ | 96 | 1 | - | - | - | - |
| $y_{2a}$ | $y_2 + z_2$ | 96 | Avg6 | - | - | yes | Leak 0.2 |
| $z_{2a}$ | $z_2$ | 96 | Avg6 | - | - | yes | Leak 0.2 |
| $y_\mu$ | $y_{2a}$ | k | 1 | 1 | - | - | - |
| $y_\sigma$ | $y_{2a}$ | k | 1 | 1 | - | - | ExpN |
| **y** | $y_\mu + y_\sigma$ | k | - | - | - | - | Softmax |
| $z_\mu$ | $z_{2a}$ | 64 | 1 | 1 | - | - | - |
| $z_\sigma$ | $z_{2a}$ | 64 | 1 | 1 | - | - | ExpN |
| **z** | $z_\mu + z_\sigma$ | 64 | - | - | - | - | - |

Table 8: SVHN: $G_z(y)$

| Name | Input | Channels | Width | Stride | Dropout | BatchNorm | Activation |
|------|-------|----------|-------|--------|---------|-----------|------------|
| **y** | - | k | - | - | - | - | - |
| $z_\mu$ | **y** | 64 | 1 | 1 | - | - | - |
| $z_\sigma$ | **y** | 64 | 1 | 1 | - | - | ExpN |
| **z** | $z_\mu + z_\sigma$ | 64 | - | - | - | - | - |

Table 9: SVHN: $G_x(y, G_z(y))$

| Name | Input | Channels | Width | Stride | Dropout | BatchNorm | Activation |
|---|---|---|---|---|---|---|---|
| $y$ | - | k | - | - | - | - | - |
| $z$ | - | 64 | - | - | - | - | - |
| $y_1$ | $y$ | 512 | 1 | 1 | - | - | - |
| $z_1$ | $z$ | 512 | 1 | 1 | - | - | - |
| $y_{1a}$ | $y_1 + z_1$ | 512 | - | - | - | yes | - |
| $z_{1a}$ | $z_1$ | 512 | - | - | - | yes | Leak 0.2 |
| $y_2$ | $y_{1a}$ | 256 | 4 | 1 | - | - | - |
| $z_2$ | $z_{1a}$ | 256 | 4 | 1 | - | - | - |
| $y_{2a}$ | $y_2 + z_2$ | 256 | - | - | - | yes | - |
| $z_{2a}$ | $z_2$ | 256 | - | - | - | yes | Leak 0.2 |
| $y_3$ | $y_{2a}$ | 256 | 3 | 1 | - | - | - |
| $z_3$ | $z_{2a}$ | 256 | 3 | 1 | - | - | - |
| $y_{3a}$ | $y_3 + z_3$ | 256 | - | - | - | yes | - |
| $z_{3a}$ | $z_3$ | 256 | - | - | - | yes | Leak 0.2 |
| $y_4$ | $y_{3a}$ | 128 | 3 | 2 | - | - | - |
| $z_4$ | $z_{3a}$ | 128 | 3 | 2 | - | - | - |
| $y_{4a}$ | $y_4 + z_4$ | 128 | - | - | - | yes | - |
| $z_{4a}$ | $z_4$ | 128 | - | - | - | yes | Leak 0.2 |
| $y_5$ | $y_{4a}$ | 128 | 3 | 1 | - | - | - |
| $z_5$ | $z_{4a}$ | 128 | 3 | 1 | - | - | - |
| $y_{5a}$ | $y_5 + z_5$ | 128 | - | - | - | yes | - |
| $z_{5a}$ | $z_5$ | 128 | - | - | - | yes | Leak 0.2 |
| $y_6$ | $y_{5a}$ | 4 | 2 | 1 | - | - | - |
| $z_6$ | $z_{5a}$ | 4 | 2 | 1 | - | - | - |
| $x$ | $y_6 + z_6$ | 3 | - | - | - | - | Sigmoid |

Table 10: SVHN: $D(x, y, z)$

| Name | Input | Channels | Width | Pooling | Dropout | BatchNorm | Activation |
|---|---|---|---|---|---|---|---|
| $x$ | - | 1 | - | - | - | - | - |
| $y$ | - | k | - | - | - | - | - |
| $z$ | - | 64 | - | - | - | - | - |
| $x_1$ | $x$ | 96 | 3 | - | - | - | Leak 0.2 |
| $x_2$ | $x_1$ | 96 | 3 | - | - | - | Leak 0.2 |
| $x_3$ | $x_2$ | 96 | 3 | Max2 | - | - | Leak 0.2 |
| $x_4$ | $x_3$ | 192 | 3 | - | 0.1 | - | Leak 0.2 |
| $x_5$ | $x_4$ | 192 | 3 | - | - | - | Leak 0.2 |
| $x_6$ | $x_5$ | 192 | 3 | Max2 | - | - | Leak 0.2 |
| $x_7$ | $x_6$ | 384 | 3 | - | 0.1 | - | Leak 0.2 |
| $x_8$ | $x_7$ | 192 | 1 | - | - | - | Leak 0.2 |
| $x_9$ | $x_8$ | 96 | 1 | Avg6 | - | - | Leak 0.2 |
| $y_1$ | $y$ | 96 | 1 | 1 | - | - | Leak 0.2 |
| $y_2$ | $y_1$ | 96 | 1 | 1 | 0.1 | - | Leak 0.2 |
| $z_1$ | $z$ | 96 | 1 | 1 | - | - | Leak 0.2 |
| $z_2$ | $z_1$ | 96 | 1 | 1 | 0.1 | - | Leak 0.2 |
| $p_0$ | $x_5 \mid y_2 \mid z_2$ | 288 | - | - | - | - | - |
| $p_1$ | $p_0$ | 288 | 1 | 1 | 0.1 | - | Leak 0.2 |
| $p_2$ | $p_1$ | 288 | 1 | 1 | 0.1 | - | Leak 0.2 |
| $p$ | $p_1$ | 1 | 1 | 1 | 0.1 | - | - |

