# OpenReview forum: "Adversarially Learned Mixture Model"
_ICLR.cc/2019/Conference_

### Official Review · AnonReviewer1 · 2018-10-27
**A GAN variant for joint discrete-continous latent model**

**Rating:** 6
**Confidence:** 2

**Review:**

The paper presents a generative model that can be used for unsupervised and semi-supervised data clustering. unlike most of previous method  the latent  variable is composed of both continuous and discrete variables. Unlike previous methods like ALI the conditional probability  p(y|x) of the labels given the object is represented by a neural network and not simply drown from the data.  The  authors show a  clustering error rate on the MNIST data that is better than previously proposed methods.

---

### Official Review · AnonReviewer3 · 2018-10-28
**the writing should be improved.**

**Rating:** 5
**Confidence:** 4

**Review:**

The paper presents a method for un-/semi- supervised clustering which combines adversarial learning and Mixture of Gaussian.

Cons:
It is interesting to incorporate the generative model into GAN.

Probs:
1.	The author claims the method is the first one to generative model inferring both continuous and categorical latent variables. I think that such a conclusion is overclaimed, there are a lot of related works, e.g., Variational deep embedding: An unsupervised generative approach to Clustering, IJCAI17; Adversarial Variational Bayes: Unifying Variational Autoencoders and Generative Adversarial Networks, ICML2017. Multi-Modal Generative Adversarial Networks for Diverse Datasets, ICLR19 submission. In fact, these methods are also very competitive approaches to clustering.
2.	Is adversarial learning do helpful for clustering with the generative model? Some theoretical should be given, at least, some specified experiments should be designed.
3.	The novelty should be further summarized by highlighting the difference with most related works including but not limited the aforementioned ones. The current manuscript makes the work seem like a straightforward combination of many existing approaches.
4.	In fact, the paper is hard to follow. I would recommend improving the logic/clarity.

---

### Official Review · AnonReviewer2 · 2018-11-01
**ADVERSARIALLY LEARNED MIXTURE MODEL**

**Rating:** 6
**Confidence:** 1

**Review:**

The paper uses Generative Adversarial Networks (GAN) for unsupervised and semi-supervised clustering. Neural network based generators are used for sampling using a mixture model. The parameters of the generators are optimised during training against a discriminator that tries to distinguish between generated distributions. Experimental results on MNIST and SVHN datasets are given to motivate their models.

I am far from being an expert on GANs but just from a clustering viewpoint I can make the following comments about the paper:
- Comparison with other clustering techniques is not present. How does error and computational efficiency compare with other techniques?
- There doesn’t seem to be any deep theoretical insight. This paper is more about using GANs in a particular way (different than the previously attempted ways) to study and demonstrate results. Once the model is decided so are the algorithms.
- I am not sure what is the standard practice of describing the algorithms in the context of GANs. I found parsing Appendix A and B very difficult.

---

### Public Comment · (anonymous) · 2018-12-10
**Implementation details and code**

This is an interesting paper. We tried to implement it but can not achieve the same performance in the paper. The margin is large. Could the author help to provide some implementation details or share the code?

---

### Meta-Review · Area_Chair1 · 2018-12-15

**Confidence:** 3
**Recommendation:** Reject

**Metareview:**

The paper presents a method for unsupervised/semi-supervised clustering, combining adversarial learning and the Mixture of Gaussians model. The authors follow the methodology of ALI, extending the Q and P models with discrete variables, in such a way that the latent space in the P model comprises a mixture-of-Gaussians model.

The problem of generative modeling and semi-supervised learning are interesting topics for the ICLR community.

The reviewers think that the novelty of the method is unclear. The technique appears to be a mix of various pre-existing techniques, combined with a novel choice of model. The experimental results are somewhat promising, and it is encouraging to see that good generative model results are consistent with improved semi-supervised classification results. The paper seems to rely heavily on empirical results, but they are difficult to verify without published source code. The datasets chosen for experimental validation are also quite limited, making it it difficult to assess the strengths of the proposed method.